# INVARIANT SPATIOTEMPORAL REPRESENTATION LEARNING FOR CROSS-PATIENT SEIZURE CLASSIFICATION

## ABSTRACT

Automatic seizure type classification from electroencephalogram (EEG) data can help clinicians to better diagnose epilepsy. Although many previous studies have focused on the classification problem of seizure EEG data, most of these methods require that there is no distribution shift between training data and test data, which greatly limits the applicability in real-world scenarios. In this paper, we propose an invariant spatiotemporal representation learning method for cross-patient seizure classification. Specifically, we first split the spatiotemporal EEG data into different environments based on heterogeneous risk minimization to reflect the spurious correlations. We then learn invariant spatiotemporal representations and train the seizure classification model based on the learned representations to achieve accurate seizure-type classification across various environments. The experiments are conducted on the largest public EEG dataset, the Temple University Hospital Seizure Corpus (TUSZ) dataset, and the experimental results demonstrate the effectiveness of our method.

## 1 INTRODUCTION

Epilepsy is a pervasive neurological disease that affects individuals all over the world, with considerable cognitive, psychological, and social ramifications (Beghi, 2020). The mainstream approach to epilepsy diagnosis relies on EEG data to classify seizures (Falco-Walter, 2020; Fisher et al., 2017). However, traditional methods based on human labor are not only costly but also susceptible to human uncertainty, as these methods require clinicians to meticulously review extensive EEG recordings (Jiang et al., 2017). As a result, using machine learning techniques to automatically classify seizure types attracts increasing attention.

Current seizure classification scenarios can be divided into two categories, as illustrated in Figure 1. The first category is patient-specific, with a consistent distribution between the training and test sets (Yuan et al., 2023; Rout et al., 2022). However, a recent study (Karimi-Rouzbahani & McGonigal, 2024) highlights the necessity of developing cross-patient classifications due to the significant variability in EEG patterns across individuals, such as differences in epileptogenic zones and brain structure. Meanwhile, another study (Jirsa et al., 2014) also demonstrates that the electrophysiological signatures of seizures can differ significantly across patients due to individual variations in brain connectivity and the mechanisms underlying seizure generation, underscoring the need for developing cross-patient seizure prediction models.

Several approaches have been explored in the field of EEG-based seizure classification, aiming to improve the accuracy and generalizability of identifying seizure patterns. Early machine learning methods for accurately classifying EEG data included Support Vector Machines (SVM), k-Nearest Neighbors (k-NN), and Bayesian methods (Lazcano-Herrera et al., 2021; Sha'Abani et al., 2020). With the advancement of deep learning, further methods have been explored. Convolutional Neural Networks (CNNs) (Supriya et al., 2021; Craik et al., 2019) apply convolution methods to efficiently learn spatiotemporal feature representations of EEG signals. In parallel, Recurrent Neural Networks (RNNs) (Huang et al., 2019; Shoeibi et al., 2021; Ma et al., 2023), were employed to leverage their capacity for capturing temporal dependencies and dynamics.

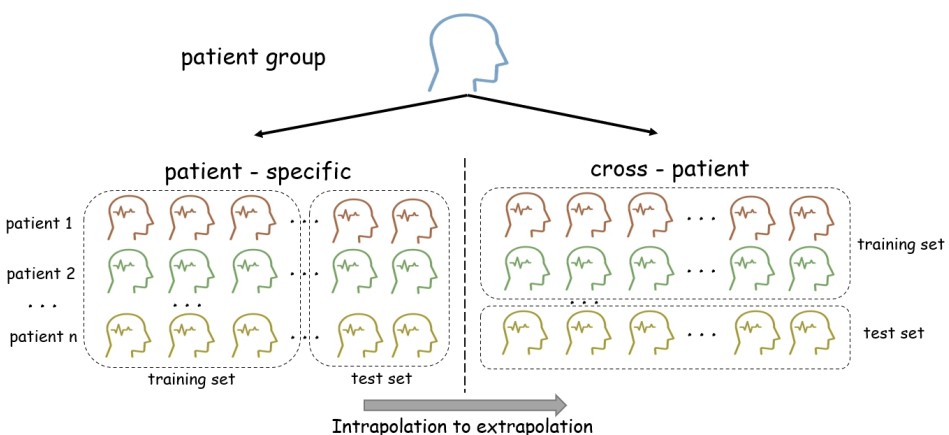

Figure 1: Two seizure classification scenarios.

To address non-Euclidean geometric properties overlooked by CNNs and RNNs, Graph Neural Networks (GNNs) have been proposed, to model the spatial relationships between EEG electrodes using a graph representation (Hajisafi et al., 2023; He et al., 2022; Klepl et al., 2024). However, these methods face inherent limitations in generalization performance. Moreover, recent approaches (Zhang et al., 2020) for dealing with the cross-patient problem struggle to implement effective adversarial learning when applied to larger and more diverse patient groups. Consequently, these challenges in efficiently addressing the cross-patient problem remain unsolved.

In order to address previous deficiencies, we proposed a novel spatiotemporal invariant risk minimization loss to solve this problem. Specifically, we first use the invariant mask function to separate the raw EEG feature into the invariant representation and variant representation and use self-supervised learning (SSL) to guarantee the preserved invariant information is able to predict the invariant feature at the next timestamp. In addition, we use the label information to generate the supervised signal to ensure the preserved invariant information can predict the seizure type. Finally, we use the variance of the gradient toward the mask function to control the time-varying variation of our methods in different patient groups.

We highlight our contributions as follows:

- We use the mask function to capture the invariant spatiotemporal information in the raw EEG data and use such information for self-supervised learning.

- To further control the variation of the loss of the classification model, we use the variance of the gradient as the penalty to achieve invariant learning.

- The experiments on the largest public dataset verify the effectiveness of our method.

## 2 RELATED WORK

### 2.1 EEG DATA CLASSIFICATION

Electroencephalography (EEG) data classification involves processing and analyzing EEG signals to identify and distinguish different brain activity patterns or states, thereby enabling the classification and diagnosis of brain functions and conditions (Rabcan et al., 2020). Early machine learning methods for accurately classifying EEG data included Support Vector Machines (SVM), k-Nearest Neighbors (k-NN), and Bayesian methods (Sha'Abani et al., 2020; Lazcano-Herrera et al., 2021). With the rapid development of deep learning, recent EEG classification methods can be broadly categorized into those based on Convolutional Neural Networks (CNN), Recurrent Neural Networks (RNN), and Graph Neural Networks (GNN) (Klepl et al., 2024).

The core idea of CNN-based methods is to automatically learn spatiotemporal feature representations of EEG signals through convolutional operations, effectively identifying and classifying brain

signals (Craik et al., 2019). In recent advances, EEG-DBNet decodes the temporal and spectral sequences of EEG signals using two parallel network branches, each containing local and global convolution blocks to extract local and global features (Lou et al., 2024). ACPA-ResNet enhances the model's ability to identify key features by introducing attention mechanisms and fully pre-activated residual blocks (Yutian et al., 2024). To improve classification efficiency, EDPNet employs a lightweight adaptive time-frequency fusion module to integrate time-frequency information from multiple electrodes and uses a parameter-free multi-scale variance pooling module to extract more comprehensive temporal features (Han et al., 2024). While CNNs have proven they outperform in capturing spatiotemporal features, RNNs are also employed for their efficiency in capturing temporal dependencies and dynamics over time. Ma et al. (2023) proposed a model that combines CNN for spatial feature extraction with a Bi-LSTM network to effectively capture the temporal dynamics of EEG signals.

The core idea of GNN-based methods is to capture non-Euclidean geometric properties by modeling the spatial relationships between EEG electrodes (Klepl et al., 2024). REST combines dynamic graph neural networks and recurrent structures, achieving efficient EEG data classification through a residual state update mechanism (Afzal et al., 2024). NeuroGNN improves classification accuracy by capturing the dynamic interactions between EEG electrode positions and the corresponding brain regions' semantics within a dynamic graph neural network framework (Hajisafi et al., 2024). Tang et al. (2022) proposed a method that combines self-supervised pre-training and GNN, constructing a graph structure of spatial and dynamic brain connectivities between EEG electrodes and processing spatiotemporal dependencies using a Diffusion Convolutional Recurrent Neural Network (DCRNN) (Tang et al., 2022).

## 2.2 EXTRAPOLATION IN MEDICAL DATA

Extrapolation in medical data, as well as the cross-patient problem, refers to the divergence between test and training data distributions, which may be attributed to the spatial-temporal evolution of patient data (Zhang et al., 2024b). This distribution shift could be partly attributed to the spatial-temporal evolution of data (Zhang et al., 2022; Liu et al., 2021b). Previous studies can be broadly categorized into three types to address this problem.

The first category focuses on representation learning, particularly unsupervised methods that aim to generate domain-agnostic features (Zhang et al., 2020; Yang et al., 2022; 2021). By leveraging either domain generalization or expert-guided structuring of features, these approaches aim to enhance capacity of the model to generalize to new distributions by capturing essential patterns in the data. This ensures that the learned representations retain attributes conducive to better performance across unseen domains. The second category revolves around supervised models designed to enhance generalization by employing techniques such as causal learning and invariant risk minimization. These approaches emphasize end-to-end learning strategies, which have been shown to improve robustness to distributional shifts (Parulekar et al., 2023; Oberst et al., 2021; Mazaheri et al., 2023). The third category involves optimization-based approaches, including distributionally robust optimization (DRO), which focuses on minimizing the worst-case performance under potential shifts in the data (Sagawa et al., 2019; Liu et al., 2021a).

## 3 PRELIMINARY

### 3.1 PROBLEM SETUP

The primary objective of the seizure classification task is to predict the seizure type from a given EEG signal clip. These clips were sliced from seizure EEGs using non-overlapping sliding windows with fixed temporal size $T$. For each sample, we denote $X \in \mathbb{R}^{T \times N \times M}$ as the EEG clip feature after preprocessing, where $T$ is the temporal length of the EEG clip, $N$ is the number of EEG channels/electrodes, and $M$ is the number of features obtained through Fast Fourier Transform (FFT). Meanwhile, we denote $y$ as the seizure class label. For the independent identical distributed scenario, different clips from the same patient may appear in both the training and test sets. However, in real healthcare scenarios, patients in the test sets (a group of new patients) may completely unseen in the training set, leading to the cross-patient problem (Zhang et al., 2024a), which can be further

formulated as follows: The patient set $P$ is divided into two disjoint subsets, $P_T$ and $P_D$, such that $P_T \cup P_D = P$ and $P_T \cap P_D = \emptyset$. Here, $P_D$ is used for training, and $P_T$ is used for testing.

## 3.2 PREVIOUS GRAPH-BASED METHODS FOR EEGS

**Graph Representing.** Let $\mathcal{G} = \{\mathcal{V}, \mathcal{E}, \boldsymbol{W}\}$ denote the graph structure, where $\mathcal{V}$ is the set of nodes, $\mathcal{E}$ refers to the set of edge, and $\boldsymbol{W}$ is the adjacency matrix of the graph. With EEGs, $\mathcal{V}$ often denotes the electrodes, and $E$ represents if two electrodes are connected, the adjacent matrix depicts the connection strength among these electrodes. In consideration of the distribution of nodes and the physiological properties of the brain, two distinct approaches to graph construction on EEG clips are evident. One undirected distance graph-based approach is to utilize the Euclidean distance between different nodes on standard 10-20 EEG electrode placement as the basis, followed by the threshold Gaussian kernel to determine the weights between $v_i$ and $v_j$ ($v_i, v_j \in \mathcal{V}$):

$$W_{ij} = \begin{cases} \exp\left(-\dfrac{\text{dist}(v_i, v_j)^2}{\sigma^2}\right) & \text{if } \text{dist}(v_i, v_j) \leq \kappa \\ 0 & \text{otherwise,} \end{cases}$$

where $\text{dist}(v_i, v_j)$ represents the Euclidean distance between two nodes $v_i$ and $v_j$, $\sigma$ is the standard deviation of the distances, while $\kappa$ is the threshold for sparsity.

An alternative approach, based on a directed correlation graph, particularly focuses on the dynamic connectivity between different nodes. To evaluate the connectivity, only the weights that fall within the most $k$ similar neighbors (including self-edges) are preserved to ensure the sparsity of the graph. The weight can be formulated as follows:

$$W_{ij} = \begin{cases} Corr(\mathbf{X}_{:,i,:}, \mathbf{X}_{:,j,:}) & \text{if } v_j \in \mathcal{C}_k(v_i) \\ 0 & \text{otherwise,} \end{cases}$$

where $X_{:,i,:}$ and $X_{:,j,:}$ denotes the preprocessed signals in $v_i$ and $v_j$, $Corr(\cdot, \cdot)$ represents the pearson correlation coefficient, and $\mathcal{C}_k(v_i)$ referring to the most $k$ similar neighbors of $v_i$.

**Diffusion Convolutional Recurrent Neural Network.** Previous works utilize the diffusion convolutional recurrent neural network (DCRNN) to effectively capture the temporal and spatial dependencies in EEG signals. To capture the temporal dependencies in EEG data, modified gated recurrent units (GRUs) (Cho et al., 2014) are employed.

For spatial dependency, diffusion convolution provides significant insights into the influence exerted by each node on all others, and the extent of this kind of influence can be quantified by applying a bidirectional random walk on the directed graph and calculating a $K$-step diffusion convolution. The diffusion convolution is defined by:

$$X_{:,m \star \mathcal{G}} f_\theta = \sum_{k=0}^{K-1} \left(\theta_{k,1}(D_O^{-1}W)^k + \theta_{k,2}(D_I^{-1}W^\mathsf{T})^k\right) X_{:,m}, \quad m \in \{1, \ldots, M\},$$

where $X$ is the preprocessed segment with $N$ nodes and $M$ features at timestamps $t \in \{1, \cdots, T\}$, $\theta \in \mathbb{R}^{K \times 2}$ are the parameters of the filter, and $D_O$ and $D_I$ are the out-degree and in-degree diagonal matrices of the graph. The transition matrices for the diffusion processes are $D_O^{-1}W$ and $D_I^{-1}W^\mathsf{T}$.

For undirected graphs, the process converts to ChebNet spectral graph convolution (Defferrard et al., 2016), where $X_{:,m}$ is filtered using Chebyshev polynomial bases. The spectral graph convolution can be expressed as

$$X_{:,m \star \mathcal{G}} f_\theta = \boldsymbol{\Phi} \left(\sum_{k=0}^{K-1} \theta_k \boldsymbol{\Lambda}^k\right) \boldsymbol{\Phi}^\mathsf{T} X_{:,m}, \quad m \in \{1, \ldots, M\},$$

where $\boldsymbol{\Phi}$ and $\boldsymbol{\Lambda}$ are the eigenvector and eigenvalue matrices of the graph Laplacian $\mathbf{L}$. This is equivalent to

$$X_{:,m \star \mathcal{G}} f_\theta = \sum_{k=0}^{K-1} \theta_k \mathbf{L}^k X_{:,m},$$

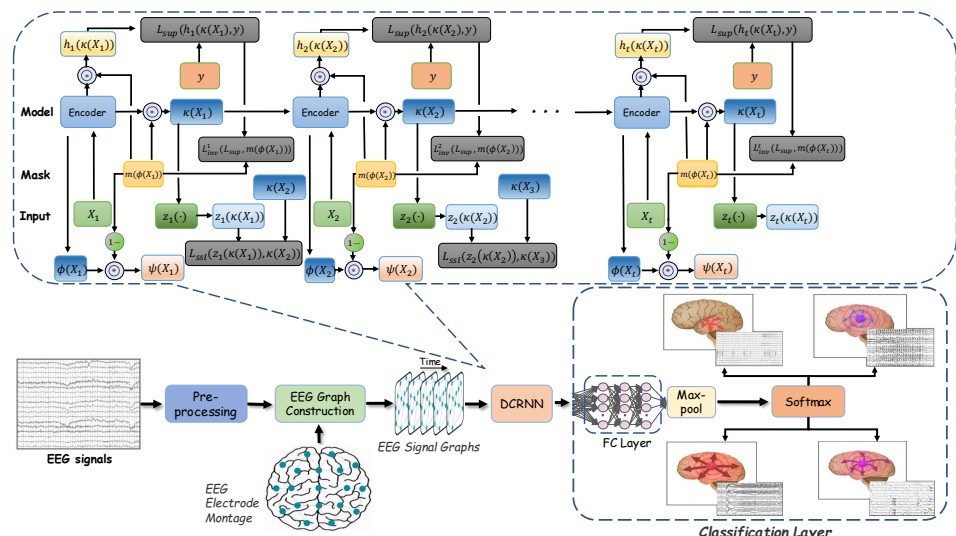

Figure 2: Overview of the proposed spatiotemporal invariant learning method.

and can be further approximated using Chebyshev polynomials as

$$X_{:,m \star \mathcal{G}} f_\theta = \sum_{k=0}^{K-1} \tilde{\theta}_k T_k(\tilde{L}) X_{:,m}, \quad m \in \{1, \dots, M\},$$

where $T_k(\tilde{L})$ is the $k$-th Chebyshev polynomial of the scaled Laplacian $\tilde{L}$, allowing for efficient computation without explicit eigenvalue decomposition.

## 4 METHODOLOGY

In a cross-patient scenario, we propose the spatiotemporal invariant risk minimization (ST-IRM) loss, making the prediction model achieves both (a) accurately predicting patient's seizure type in each patient group; (b) The variation of prediction between the different groups is small. Specifically, for a timestamp $t$, we derive an invariant mask function $m(\cdot)$ to separate the representations of the raw EEG feature into two orthogonal components. We denote the representation of the raw EEG feature as $\phi(X_{:,:,t})$. For simplification of notations, we use $X_t$ instead of $X_{:,:,t}$. The representation in the present paper is obtained by DRCNN. Through the invariant mask function $m(\cdot)$, $\phi(X_t)$ is decomposed into an invariant representation $\kappa(X_t) = m(\phi(X_t))$, and the variant representation $\psi(X_t) = (1 - m(\phi(X_t))) \odot \phi(X_t)$, where $m(X_t) \in [0,1]^{N \times M}$. For example, the invariant representation for an EEG signal data includes the key signals that determine the seizure type of the patient; while the variant representation records the noise and artifact information such as the blinking, muscle movement of the patient or the measure error of the signal detection machines. The decomposition helps us to recognize the components which play the causal role in discriminating the seizure type, and the non-causal features that would vary across patients. Utilizing the non-causal features will help us get a better classification of known patients in the training set, but the unknown patients in the test set would possess different features. The utilization of these features would disturb the classifier and harm the generalization of the seizure classifier to unknown patients. Thus, it is important to conduct the decomposition. Next, we introduce our method in detail step by step.

In time-series data, especially in the EEG data, there should be some correlation of the previous representations $X_{t-1}$ with the current feature $X_t$ (Tang et al., 2022). Unlike the previous SSL approach that aims to learn a model $z_t(\cdot)$ to ensure $z_{t-1}(X_{t-1}) \approx X_t$, we claim that preserve the relation between the variant parts, $\psi(X_{t-1})$ and $\psi(X_t)$ may not be helpful due to the spurious correlation. We expect only a good prediction performance between the invariant representations.

Thus, the proposed SSL loss is as below:

$$\mathcal{L}_{ssl} = \frac{1}{|nT|} \sum_{i=1}^{n} \sum_{t=1}^{T} \mathcal{L}(z_{t-1}(m(\phi(X_{t-1}^i))), m(\phi(X_t^i))),$$

where $\mathcal{L}(\cdot, \cdot)$ is the loss function such as mean-square-error loss and $X_t^i \in \mathbb{R}^{N \times M}$ is the preprocessed signal for sample $i$ at timestamp $t$. In addition, we want the information preserved by the mask function can not only predict the next invariant representation but also can predict the final seizure type, thus we use the following loss to provide the supervised signal for training the mask function:

$$\mathcal{L}_{sup} = \frac{1}{|n|} \sum_{i=1}^{n} \mathcal{L}(h_T(m(\phi(X_T^i))), y_i),$$

where $h_T(\cdot)$ is the classification model and $y_i$ is the ground truth label. We only use the representation at the last timestamp of an EGG clip to predict the seizure type. It is because we believe the representation of the last timestamp contains the information of previous timestamps given the assumption of our SSL approach and the temporal continuity nature of the EGG.

In addition, an ideal mask function $m(\cdot)$ should be able to capture the invariant representation from the raw EGG data. The conventional invariant risk minimization approach realizes this goal by setting a series of environments and learning a predictor that performs consistently well across these environments. We set the environment in the present study by partitioning the patients into groups. Since each group consists of exclusive members of patients, it naturally leads to a completely distinguished environment. To make these environments more separable, we use the clustering methods, of which the K-means is a representative, to partition the patients and the preprocessed EGG clips. The clustering method separates the samples into groups where within the group, they share similar characteristics while the samples in two different groups also possess distinguished characteristics. We construct the environments in this way to ensure difference environments share minimal commonness. Thus, a classifier that performs consistently across these environments would truly learn the invariant components and suffer the least from spurious correlations. Assuming there is a total of $G$ groups/environments, and the group indicator of each sample is denoted by $g_i$. The supervised loss at timestamp $t$ for the group $g$ is given by

$$\mathcal{L}_{sup}^{g,t} = \frac{1}{\sharp\{i : g_i = g\}} \sum_{\{i:g_i=g\}} \mathcal{L}(h_t(m(\phi(X_t^i))), y_i),$$

where $\sharp$ denotes the cardinal number of the set. It represents the supervised loss within the $g$-th group. Combining the group-based supervised loss, the overall invariant risk loss at timestamp $t$ is composed of two major terms:

$$\mathcal{L}_{inv}^t = \mathbb{E}_{g \in \mathcal{G}} \mathcal{L}_{sup}^{g,t} + \lambda \left\| \mathrm{Var}_{g \in \mathcal{G}} \left( \nabla_{\Theta^m} \mathcal{L}_{sup}^{g,t} \odot m(\phi(X_t)) \right) \right\|^2,$$

where $\Theta^m$ is the parameter of the mask function, and $\lambda$ is the hyper parameter for tuning. The previous term can be naively computed by $\frac{1}{n} \sum_{g \in \mathcal{G}} \mathcal{L}_{sup}^{g,t}$, suggesting the overall supervised loss at timestamp $t$; while the second term penalizes the classifier to perform consistently across groups. The variance depicts the variation across the environments: the lower the variance is, the more consistent performance the classifier obtains, thus, the better invariant presentation the classifier has learned with. In the second term, we multiply the gradient with the mask function for scaling. Functions with large magnitudes of parameters tend to produce lower values of the gradients. Thus, when the parameters in the mask functions get sufficiently large, the second term without scaling would be close to zero and be useless in penalizing the loss. For further incorporating the spatiotemporal information, because the more information being observed, the more accurate classification should be, we propose the weight decay loss below:

$$\mathcal{L}_{inv} = \sum_{t=1}^{T} w^{T-t} \mathcal{L}_{inv}^t,$$

where $w \in (0, 1)$ is the weight decay rate, which is a hyper-parameter for tuning. The above loss makes full use of the loss at each timestamp. The weight decay rate guarantees the most last

Table 1: Performance comparison of different methods under 12-second and 60-second scenario.

| | 12-s | | | 60-s | | |
|---|---|---|---|---|---|---|
| Method | F1 | Recall | Precision | F1 | Recall | Precision |
| CNN-LSTM | $0.596 \pm 0.035$ | $0.654 \pm 0.030$ | $0.647 \pm 0.036$ | $0.623 \pm 0.028$ | $0.661 \pm 0.030$ | $0.647 \pm 0.036$ |
| LSTM | $0.690 \pm 0.043$ | $0.724 \pm 0.033$ | $0.725 \pm 0.041$ | $0.692 \pm 0.011$ | $0.718 \pm 0.007$ | $0.717 \pm 0.017$ |
| Dense-CNN | $0.657 \pm 0.069$ | $0.690 \pm 0.053$ | $0.694 \pm 0.049$ | $0.653 \pm 0.085$ | $0.704 \pm 0.057$ | $0.659 \pm 0.118$ |
| MSTGCN | $0.670 \pm 0.031$ | $0.719 \pm 0.023$ | $0.734 \pm 0.029$ | $0.647 \pm 0.046$ | $0.696 \pm 0.027$ | $0.694 \pm 0.030$ |
| NeuroGNN | $0.647 \pm 0.040$ | $0.710 \pm 0.024$ | $0.744 \pm 0.030$ | $0.698 \pm 0.044$ | $0.733 \pm 0.042$ | $0.714 \pm 0.056$ |
| Corr-DCRNN | $0.729 \pm 0.058$ | $0.756 \pm 0.041$ | $0.752 \pm 0.047$ | $0.672 \pm 0.038$ | $0.712 \pm 0.021$ | $0.705 \pm 0.029$ |
| Dist-DCRNN | $0.713 \pm 0.044$ | $0.735 \pm 0.043$ | $0.734 \pm 0.045$ | $0.695 \pm 0.028$ | $0.735 \pm 0.013$ | $0.738 \pm 0.021$ |
| PANN-DCRNN | $0.728 \pm 0.052$ | $0.753 \pm 0.042$ | $0.755 \pm 0.041$ | $0.684 \pm 0.023$ | $0.717 \pm 0.016$ | $0.720 \pm 0.024$ |
| ST-InvDCRNN(ours) | $\mathbf{0.748 \pm 0.038}$ | $\mathbf{0.772 \pm 0.028}$ | $\mathbf{0.764 \pm 0.043}$ | $\mathbf{0.713 \pm 0.043}$ | $\mathbf{0.741 \pm 0.024}$ | $\mathbf{0.742 \pm 0.037}$ |

loss weighs the heaviest for that the clips at the last timestamp contain the most information for classification and, thus should be put with the most weight. The final proposed ST-IRM loss is:

$$\mathcal{L}_{ST-IRM} = \mathcal{L}_{ssl} + \alpha\mathcal{L}_{sup} + \beta\mathcal{L}_{inv},$$

where $\alpha$ and $\beta$ are the hyper-parameters. It combines the self-supervised loss and supervised loss, which uses the invariant risk to ensure the classifier captures the invariant predictor and excludes the environment-dependent predictors. The parameters are trained with multi-task learnings by minimizing the synthesis ST-IRM loss. An overview of the proposed method is given in Figure 2.

## 5 EXPERIMENTS

### 5.1 EXPERIMENTAL SETTINGS

**Datasets.** Following previous works (Li et al., 2020; Sarić et al., 2020; Thuwajit et al., 2022), we utilized the Temple University Hospital EEG Seizure Corpus (TUSZ) dataset, which is the largest public dataset for our experiments. Specifically, we use the version v1.5.2 of the TUSZ dataset. The TUSZ dataset contains 5,612 EEG signals, and 3,050 annotated seizure events from over 300 patients, covering eight seizure types. The EEG signal was recorded using 19 electrodes from the standard 10-20 system (Homan et al., 1987).

**Data preprocessing and Experiment Details.** Following the preprocessing approach of Tang et al. (2022), we transform the raw EEG signals into the frequency domain, as seizures are associated with brain electrical activity in specific frequency bands (Tzallas et al., 2009). Following prior methodologies (Ahmedt-Aristizabal et al., 2020; Asif et al., 2020), EEG recordings were resampled to 200Hz and segmented into non-overlapped 60-second windows (clips). For seizure classification, only clips that contain a single type of seizure are considered. If a seizure event ends and another begins within the same clip, it is truncated and zero-padded to preserve a 60-second duration. Each 60-second clip is then segmented into 1-second intervals. The Fast Fourier Transform (FFT) algorithm is applied to each segment to obtain the logarithmic amplitudes of non-negative frequency components, as is outlined in Tang et al. (2022). Consequently, each 60-second clip is transformed into a sequence of 60 log-amplitude vectors. In addition, following recent studies on seizure type classification Ahmedt-Aristizabal et al. (2020); Asif et al. (2020); Tang et al. (2022), we use weighted F1-score as the main evaluation metric with precision and recall as well to measure the classification performance. The **F1-score** is the harmonic mean of precision and recall, providing a balanced measure for evaluating models, particularly when dealing with class imbalances. **Precision** is defined as the ratio of true positives (TP) to the sum of true positives and false positives (FP), expressed as: $P = \frac{TP}{TP+FP}$. This reflects the model's accuracy in predicting positive instances. **Recall**, on the other hand, is the ratio of true positives to the sum of true positives and false negatives (FN), calculated as: $R = \frac{TP}{TP+FN}$. It indicates the model's ability to identify all relevant positive cases. Finally, the **F1-score** is computed as the harmonic mean of precision and recall: $F1 = 2 \times \frac{P \times R}{P+R}$. See Appendix A for more experiment protocols and details.

**Baselines.** We compare our proposed method with several baselines, including: **CNN-based method: DenseCNN**, synthesizing advancements from both dense connections and deep inception architecture for efficient seizure classification (Ahmedt-Aristizabal et al., 2020). **RNN-based method: LSTM** (Hochreiter & Schmidhuber, 1997). **Hybrid approach that combines CNN and RNN: CNN-LSTM** (Ahmedt-Aristizabal et al., 2020), that fuses 2D-CNN and LSTM for improved

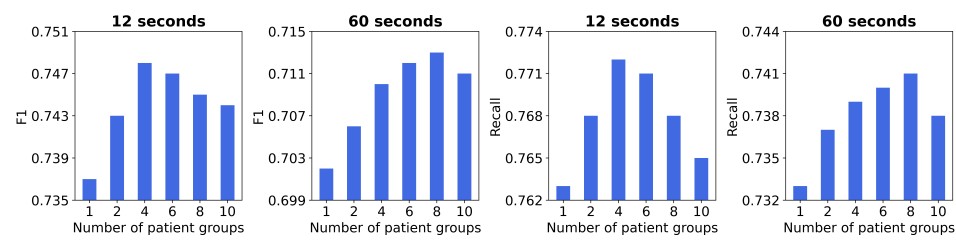

Figure 3: Performance under different numbers of patient groups.

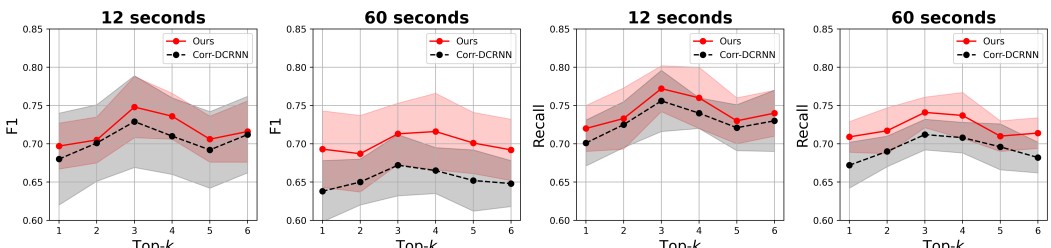

Figure 4: Performance under different values of top-$k$.

seizure classification. We also compared our method with **GNN-based methods: MSTGCN** integrates a feature extractor with a Multi-Scale Temporal Graph Convolutional Network, employing gradient reversal on patient labels to enhance cross-patient generalization capability for seizure classification (Jia et al., 2021). **Dist-DCRNN** constructs a distance graph based on Euclidean distances of the EEG node montage and applies the DCRNN model for seizure classification (Tang et al., 2022). **Corr-DCRNN** involves dynamic relations between different nodes of the brain and forms a correlation graph for the DCRNN model (Tang et al., 2022). **NeuroGNN** adopts dynamic graphs that integrate the spatial, temporal, semantic, and taxonomic properties of EEG signals to enhance seizure classification (Hajisafi et al., 2024). **PANN** employs a patient identity-focused discriminator as an adversarial optimization method to learn patient-invariant representations of EEG signals for the seizure classification task (Zhang et al., 2024a).

## 5.2 PERFORMANCE ANALYSIS

Table 1 shows the performance of our method compared with various baseline methods, evaluating with three metrics, i.e., weighted F1, Recall, and Precision scores. First, DCRNN-based models achieve competitive performance among all baselines. Second, our method significantly outperforms the baselines under both scenarios with 12-second and 60-second clip windows. Note that we adopt DCRNN as a backbone in the experiment, which is shown in Figure 2, and the superior against DCRNN-based methods demonstrates the effectiveness of our invariant learning framework.

## 5.3 IN-DEPTH ANALYSIS

To comprehensively evaluate the proposed invariant learning method, We conduct three in-depth analysis on the number of patient groups, the value of hyper-parameter top-$k$, and the classification confusion matrix, respectively. Note that the patients are clustered into groups according to their EEG recordings, Figure 3 shows the weighted F1 and the Recall scores to evaluate the performance of our method under different numbers of patient groups, for both 12-second and 60-second clip windows. We can observe that as the number of patient groups increases, the Recall-score has a similar pattern as the weighted F1-score, achieving the highest value at 4 for the 12-second case and 8 for the 60-second case.

Figure 4 shows the weighted F1 and the Recall scores to compare the performance of our method with Corr-DCRNN under different top-$k$ values, for both 12-second and 60-second clip windows. As the value of top-$k$ ranges from 1 to 6, the trend for both weighted F1 and Recall scores is increasing until a peak at around 3, followed by a slight decrease.

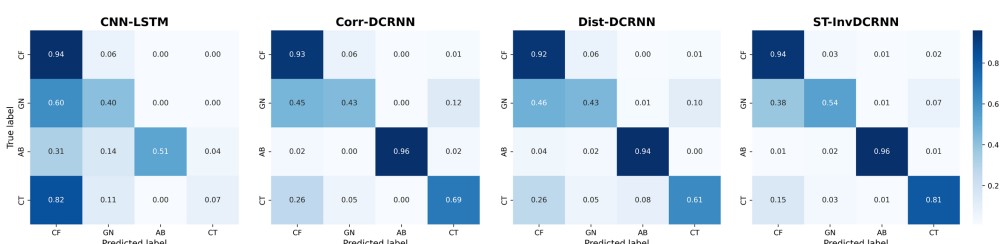

Figure 5: Confusion matrices for four classes of seizures.

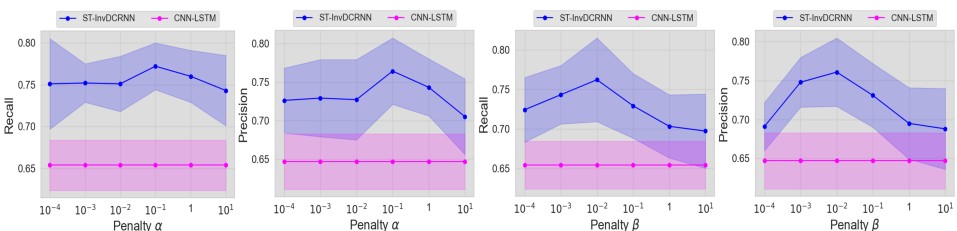

Figure 6: 12-second Performance under different penalty weights.

Figure 5 presents the confusion matrices for four seizure classification models. The comparison highlights the improved performance of our method across multiple seizure classes. In particular, the ST-InvDCRNN demonstrates superior accuracy in distinguishing between different seizure types, providing more distinct class separations, with fewer misclassifications compared to other models. A notable example is its performance in identifying the CT class, where it achieves an impressive 0.81 accuracy. This significantly surpasses the results of other methods, which tend to exhibit higher levels of confusion, especially when differentiating between CF and CT. Besides, our method achieves an accuracy of 0.54 in classifying GN seizures, significantly outperforming the baseline models, which only reach 0.40 (CNN-LSTM), 0.43 (Corr-DCRNN), and 0.43 (Dist-DCRNN). Our method shows a marked reduction in confusion between these classes, thereby providing more reliable and accurate classification. These results demonstrate the effectiveness of the ST-InvDCRNN in handling complex seizure types where other methods struggle.

Figure 6 compares ST-InvDCRNN and CNN-LSTM performance across different penalty parameters ($\alpha$ and $\beta$) for recall and precision. ST-InvDCRNN consistently outperforms CNN-LSTM, especially at intermediate penalty values. For Penalty $\alpha$, ST-InvDCRNN peaks at $\alpha = 10^{-1}$, achieving 0.772 recall score and 0.764 precision score, while CNN-LSTM shows lower scores. Similarly, for Penalty $\beta$, ST-InvDCRNN reaches its best performance at $\beta = 10^{-1}$, with 0.762 recall score and 0.761 prescision score. Overall, ST-InvDCRNN delivers better classification results than CNN-LSTM.

## 6 CONCLUSION

Epilepsy remains a significant global health challenge, with traditional EEG-based diagnostic methods posing limitations due to their reliance on clinician review. With the recent advancement of deep learning, techniques such as CNNs, RNNs, and GNNs are proposed to automatically classify the seizure type. However, existing methods often lack cross-patient robustness and guarantee, which is very common in practice. In addition, most of the methods addressing the cross-patient problem ignore the spatiotemporal information. To bridge this gap, we propose a spatiotemporal invariant risk minimization approach that addresses these challenges by adopting self-supervised learning and capturing time-varying invariant features. Experimental results on the largest public dataset verify the effectiveness of our approach, demonstrating its potential to advance epilepsy diagnosis in the cross-patient scenario. One of the possible limitations is to investigate a more efficient way to learn the model parameters and reduce the complexity while maintaining the classification performance.

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

# APPENDIX

## A EXPERIMENTAL DETAILS

Following previous works, we divide the clips and patients of the TUSZ dataset into training, validation, and test sets. The number of EEG clips is 1,925, 450, and 521 for the three sets respectively, while the number of patients is 179, 22, and 34. Note that the patient sets are disjoint for training, validation, and test sets to study the cross-patient seizure classification robustness.

We tune the following hyper-parameters on the validation set.

- $lr\_init \in [1e-5, 5e-3]$, the initial learning rate;
- top-$k \in \{1, 2, 3, 4, 5, 6\}$, the number of neighbors included in the correlation graphs for each node;
- $K \in \{2, 3, 4\}$, the maximum diffusion step;
- $d \in [0, 0.7]$, the dropout probability in the prediction networks.
- $e \in [20, 40, 60, 80, 100]$, the number of training epochs.

During the training, each batch has 40 EEG clips and the cosine annealing learning rate scheduler (Loshchilov & Hutter, 2016) is adopted. Our experiments are conducted on a computing platform of NVIDIA GeForce RTX 3090 and Intel(R) Xeon(R) Gold 6248R CPU @ 3.00GHz.

## B REPRODUCIBILITY STATEMENT

The Temple University Hospital EEG Seizure Corpus used in our study is publicly available.

Upon acceptance of this paper, the implementation code used in this work will be made publicly available to ensure reproducibility and to facilitate further research in the field.

