# OpenReview forum: "Invariant Spatiotemporal Representation Learning for Cross-patient Seizure Classification"
_ICLR.cc/2025/Conference — Submitted to ICLR 2025_

### Official Review · Reviewer_ZExC · 2024-10-28

**Soundness:** 2
**Presentation:** 2
**Contribution:** 2
**Rating:** 3
**Confidence:** 5

**Summary:**

This paper focuses on applying invariant risk minimization to subject-independent seizure type classification. The objective is to learn invariant seizure-related features across different patients, thereby mitigating the impact of subject-specific noise. The approach integrates three types of losses: self-supervised loss, supervised loss, and invariant loss, to address domain shift issues inherent in subject-independent setups for EEG classification. The self-supervised learning component enhances the robustness of the invariant representation, the supervised learning component is responsible for predicting the labels, and the invariant risk minimization component reduces domain shifts caused by the presence of multiple subjects. A Diffusion Convolutional Recurrent Neural Network (DCRNN) serves as the backbone model for training, and the Temple University Hospital EEG Seizure Corpus (TUSZ) dataset is employed for performance evaluation.

**Strengths:**

The motivation of the paper is good. It is exciting to see more researchers try to address the challenge of subject-independent classification in the EEG domain.

**Weaknesses:**

1) **Effectiveness of Invariant Risk Minimization (IRM):** Although the motivation for using IRM is sound, I personally question the actual improvement this module provides. Based on my experience with other EEG datasets in subject-independent disease diagnosis tasks, IRM has shown poor performance, often yielding little to no improvement. In this context, an ablation study examining the impact of each loss function should be included, such as evaluating model performance after completely removing the IRM loss. Additionally, clustering subjects using K-means to apply IRM does not seem reasonable. Each subject should be treated as a distinct domain since each has unique subject-specific features. This can be easily validated by performing a subject ID discrimination task (mixing samples from all subjects and using subject IDs as labels to classify).

2) **Specificity to Seizure Disease:** There is no specialized design tailored to seizure detection. What distinguishes seizure disorders from other EEG-based brain disease diagnoses, such as depression, Alzheimer's, or Parkinson’s disease? The proposed method appears more like a generalized approach for all EEG-based brain disease classification tasks under the subject-independent setup. Therefore, it would be more compelling to evaluate your method on additional EEG datasets to establish broader applicability.

3) **Clarity of Notations and Figures:** The current notations are confusing and tend to overcomplicate simple concepts. Additionally, Figure 2 is not informative enough. Instead of depicting the entire DCRNN pipeline (since the DCRNN design itself is not a novel contribution of this paper), it would be more helpful to provide a detailed illustration showing how the three loss modules are obtained and combined.

4) **Baseline Comparisons:** The baseline methods included are insufficient. In my experience, many seemingly complex methods designed for EEG classification fail even to outperform simpler models like TCN or vanilla Transformer. A comparison with TCN and vanilla Transformer models should be presented. Given that your samples are 1-second intervals with a 200Hz frequency, ensure that the TCN model includes at least 6 layers to achieve a sufficient receptive field.

**Questions:**

See weakness.

---

### Official Review · Reviewer_wBCg · 2024-11-01

**Soundness:** 2
**Presentation:** 3
**Contribution:** 2
**Rating:** 3
**Confidence:** 4

**Summary:**

This paper proposes a seizure classification method that addresses distribution shift or individual differences to improve classification performance. The proposed method extracts invariant feature representations across different sample domains to enhance generalization. Extensive experiments on TUSZ demonstrate the improved classification accuracy and effectiveness of the proposed method.

**Strengths:**

1) Tackling distribution shift issue and improving model generalization are important for the clinical settings.
2) Learning cross-domain invariant feature representations is reasonable, and the comparison results for Corr-DCRNN show its effectiveness.

**Weaknesses:**

1) The proposed method, including the model architecture and graph construction, is very similar to Corr-DCRNN [1]. Invariant feature learning is a common approach, and it seems the authors apply it to Corr-DCRNN without clarifying the necessity of this specific feature learning method. Overall, there are limited technical contributions.
2) There are many domain generalization methods available, an ablation study is needed.
3) A more comprehensive literature review would be helpful, especially as several papers address the issue of individual differences.
4) How are the extracted features evaluated for cross-domain invariance?
5) An interesting experiment would be to train the model on TUSZ and test it on another database.





[1] Tang et al. Self-supervised graph neural networks for improved electroencephalographic seizure analysis. ICLR, 2022

**Questions:**

Please refer to weakness.

---

### Official Review · Reviewer_Fme8 · 2024-11-01

**Soundness:** 2
**Presentation:** 2
**Contribution:** 2
**Rating:** 1
**Confidence:** 5

**Summary:**

Submitted manuscript discussed the implementation of framework for the classification of epileptic seizures across the intra subject scenario on different datasets.

**Strengths:**

1. Pictorial representation of the framework
2. Coordination of the sections
3. Results showcasing

**Weaknesses:**

1. Proper or necessary explanation of framework mathematical background is missing (can be improved further)
2. Discussed/ Proposing masked Diffusion Convolutional Neural Network is an existing or established method
3. Lack of novelty

**Questions:**

1. Authors need to prove the novelty of the proposed methods, as the masking strategy of Diffusion Convolution Neural Network is the existing method.
2. In Figure 2 and the methodology section, DCRNN is represented as DRCNN. This kind of typo error can be rechecked for more clarification.
3. Authors can discuss the masking function operation with the necessary mathematical foundations.
4. What is the 'k' operation and how does it change the orientation of the feature set of X_t? It can be discussed with a good explanation.
5. What is m operation? Is it a standardization operator or it has any other significance?
6. From Table 1, it is shown that the performance of 60-second trials is lesser than 12-second trials. In general epileptic seizure behavioral and spatial data will be more in 60 seconds. But still, it lags, authors can explain this scenario of the proposed method
7. Utilisation of the spatiotemporal features through spatiotemporal invariant risk minimization (ST-IRM) loss is still unclear. EEG data subjecting to the proposed network consists frequency domain characteristics, then how the proposed network can be utilises the spatiotemporal features.

---

### Official Review · Reviewer_vSYi · 2024-11-02

**Soundness:** 2
**Presentation:** 2
**Contribution:** 3
**Rating:** 5
**Confidence:** 4

**Summary:**

The authors propose heterogeneous risk minimization to partition spatiotemporal EEG data into different environments to reflect spurious correlations. They then learn invariant spatiotemporal representations and train the seizure classification model based on the learned representations to achieve accurate classification of seizure types across various environments. The method is validated on the largest public EEG dataset, the Temple University Hospital Seizure Corpus (TUSZ), demonstrating its effectiveness.

**Strengths:**

1. The authors use a masking function to capture invariant spatiotemporal information in the raw EEG data and leverage this information for self-supervised learning.

2. To further control the variability of the classification model's loss, the authors use the variance of the gradients as a penalty term to achieve invariant learning.

**Weaknesses:**

The experimental section primarily focuses on the TUSZ dataset and does not demonstrate the method's performance on other large epilepsy datasets (such as TUEP and TUSL), which may affect the method's generalizability and practical application value.

**Questions:**

Q1: The authors conducted experiments only on the TUSZ dataset, which limits the generalization capability of the results. There are many publicly available epilepsy datasets; therefore, I suggest that the authors validate their approach on larger and more diverse epilepsy datasets.

Q2: In the introduction, the authors' motivation for the study is not clearly articulated. They mention that this paper aims to address generalization ability and the cross-patient issue. However, as far as I know, cross-patient variability is no longer considered a significant problem. Since the proposed spatiotemporal invariant risk minimization (ST-IRM) loss function is intended to address these issues, it raises a question similar to my previous point (Q1): how can validation be assured on a single dataset?

Q3: The description of the spatiotemporal invariant risk minimization (ST-IRM) loss lacks detailed mathematical derivation and intuitive explanations, particularly regarding the specific implementation details of the masking function.

Q4: While the paper mentions several evaluation metrics, it lacks an in-depth analysis of the results, especially regarding performance differences across different patient groups and the reasons for those differences. The discussion of the results is relatively weak, and there is a lack of reflection on the potential limitations of the proposed method.

---

### Meta-Review · Area_Chair_KZpt · 2024-12-17

**Metareview:**

The distribution shift between training data and test data is an important issue that should be considered in real-world scenarios. This paper presents an invariant spatiotemporal representation learning method for cross-patient seizure classification. Variability across patients in clinical applications is critical, so it is important to learn invariant seizure-related features across different patients. The motivation of the paper is good, but there are a few serious concerns raised by reviewers. First of all, the clarity of notations and figures should be improved to increase the readability. There are lots of efforts for domain generalization and the IRM is just one of those. The effectiveness of IRM should be clarified.  Baseline comparisons should be also improved for future submissions. Therefore, the paper is not recommended for acceptance in its current form. I hope authors found the review comments informative and can improve their paper by addressing these carefully in future submissions.

**Additional Comments On Reviewer Discussion:**

There is no author response and no changes during the discussion period. All of reviewers stood by their original decisions.

---

### Decision · Program_Chairs · 2025-01-22

Reject